# In vivo clearance of nanoparticles by transcytosis across alveolar epithelial cells

Pascal Detampel[1☯¤], Anutosh Ganguly[2,3☯]*, Sara Tehranian[4], Francis Green[5], Santiswarup Singha[2,6], Pere Santamaria[2,6], Ayodeji A. Jeje[4], Clifford S. Cho[3], Björn Petri[2,6], Matthias W. Amrein[1]*

**1** Department of Cell Biology and Anatomy, University of Calgary, Calgary, Canada, **2** Department of Microbiology, Immunology and Infectious Diseases, University of Calgary, Calgary, Canada, **3** Department of Surgery, University of Michigan at Ann Arbor, Ann Arbor, Michigan, United States of America, **4** Department of Chemical and Petroleum Engineering, University of Calgary, Calgary, Canada, **5** Department Pathology & Laboratory Medicine, University of Calgary, Calgary, Canada, **6** The Snyder Institute for Chronic Diseases, University of Calgary, Calgary, Canada

☯ These authors contributed equally to this work.
¤ Current address: Division of Pharmaceutical Technology, Department of Pharmaceutical Sciences, University of Basel, Basel, Switzerland
* mamrein@ucalgary.ca (MWA); ganutosh@med.umich.edu (AG)

**Data Availability Statement:** All supplemental figures and supporting documents are submitted with the manuscript.

## Abstract

Nanoparticles in polluted air or aerosolized drug nanoparticles predominantly settle in the alveolar lung. Here, we describe a novel, highly effective pathway for the particles to cross the alveolar epithelium and reach the lymph and bloodstream. Amorphous silica nanoparticles, suspended in perfluorocarbon, were instilled into the lungs of mice for intravital microscopy. Particles formed agglomerates that settled on the alveolar wall, half of which were removed from the lung within 30 minutes. TEM histology showed agglomerates in stages of crossing the alveolar epithelium, in large compartments inside the epithelial cells and crossing the basal membrane into the interstitium. This pathway is consistent with published kinetic studies in rats and mice, using a host of (negatively) charged and polar nanoparticles.

## Introduction

Amorphous silica nanoparticles from natural sources [1] and at workplaces are common in ambient air. Upon breathing, the nanoparticles (<0.1 μm, also termed ultrafine particles) reach the alveolar region of the lung to an increasing extent with decreasing size and are mainly deposited in the alveolar region [2]. Unlike crystalline silica that can cause silicosis, chronic obstructive pulmonary disease (COPD), and lung cancer [3,4], amorphous silica is quickly cleared, potentially explaining why it has few if any pathologic effects [5]. A detailed kinetic study in rats using silica-coated as well as other radiolabeled nanoparticles reported a biphasic clearance from the lung with a $t_{1/2}$ of only 0.2–0.3 hours for the first phase and $t_{1/2}$ of 1.2 days for the second phase, leading to complete clearance [6]. Another study found a large range of nanoparticles with noncationic surface charges to be rapidly translocated from the lung lumen to the mediastinal lymph nodes [7]. The speed and efficiency of clearance for these particles are inconsistent with removal by alveolar macrophages [8] that takes weeks to years

**Funding:** This work was supported by grants from Canadian Institute for Health Research (CIHR) to MA, Nanotechnology scholarship from Alberta Ingenuity (now part of Alberta Innovates) to ST and the Swiss National Science Foundation (SNSF, PBBSP3-146963) to PD. This work was also supported by the Snyder Mouse Phenomics Resources Laboratory and Live Cell Imaging Facility, funded by the Snyder Institute for Chronic Diseases at the University of Calgary.

**Competing interests:** The authors have declared that no competing interests exist.

**Abbreviations:** AEC, alveolar epithelial cells from mice; COPD, chronic obstructive pulmonary disease; hAEC, human primary alveolar epithelial cells; MFI, mean fluorescence intensity; PFC, perfluorocarbon; TEM, transmission electron microscopy; TGN, trans-Golgi network.

to complete [9,10]. As well, particles within macrophages leave the lung via the mucociliary pathway. For alveolar particles to rapidly reach the lymph and blood implies that they first cross the epithelium.

Here we show by intravital microscopy in mice how particles settle on the alveolar wall in agglomerates and are then removed from the area at a rate consistent with the published kinetics for these and similar nanoparticles. Electron microscopy of the lungs show the agglomerates in stages of crossing the alveolar epithelium: Particles adhered to the epithelium, inside the epithelial cells in large compartments, or in the interstitium. The process appears geared towards agglomerates but also occurs for individual particles. Note that agglomerates are the predominant form for nanoparticles in aerosols and in liquids [11].

The novel pathway, described in this paper, constitutes a key advance in understanding inhalation toxicology and clearance of nanoparticles. Considering access to the transepithelial pathway may also be relevant to inhalation drug design.

## Materials and methods

### Materials

Fluorescent (red and infrared, Table 1) amorphous silica particles of 50 nm in size were purchased from Kisker Biotech (Steinfurt, Germany). Silica nanoparticles with a size of 58 nm and silver core of 28 nm were obtained from nanoComposix (San Diego, USA). Nanopowder of titanium (IV) dioxide (contains 1% Mn as dopant, nanopowder, <100 nm particle size (BET), ≥97%), sodium azide (NaN$_3$), and 2-deoxy-D-glucose were purchased from Sigma-Aldrich (Oakville, Canada), perfluorocarbon FC-43 (PFC) was obtained from 3M (London, Canada).

### Intravital imaging

We adapted a protocol [12]. for inverted lung microscopy to now include partial liquid ventilation. Female C57BL/6 mice were obtained from Jackson Laboratories (Bar Harbor, ME) and used between 8 and 16 weeks of age. The entire protocol (no. AC13-0105) was approved by the University of Calgary Animal Care Committee. Mice were anesthetized with ketamine (150 μg/g body weight) and xylazine (10 μg/g body weight) i.p and the jugular vein was cannulated for further central access. Tracheotomy was performed and a ventilation cannula (outer diameter 1.2 mm, inner diameter 0.8 mm) was inserted into the trachea and secured around

**Table 1. Nanoparticles and concentrations used *in vivo*.**

| Experiment | Particle (Abbreviation) | Diameter (nm) | Fluorescence (Ex / Em) | Conc (mg/ml) | mg/kg μg/gm | μg/ animal* |
|---|---|---|---|---|---|---|
| **Intravital** | Silica (S50-Red) | 50 | Red (569 / 585) | 0.5 | 3.8 | 76 |
| **Isolated organ Imaging** | Silica (S50-IR) | 50 | Infrared (754 / 778) | 9 | 36 | 720 |
| **Electron microscopy** | Silica w/ Ag core (S-Ag) | 58 | None | 1.0 | 3.0 | 60 |
| | Silica (S50-Red) | 50 | Red (569 / 585) | 0.95 | 3.8 | 76 |
| | Silica (S50-Red) | 50 | Red (569 / 585) | 0.05 | 0.15 | 3 |
| | TiO2 1% Mn doped (TiO2-Mn) | <100 | None | 0.5 | 3.8 | 76 |

*Average animal was 20g.

the lower front teeth with sutures. Inside the cannula, PE-10 tubing was inserted into the cannula to facilitate administration of nanoparticles, while, at the same time, ventilating the mouse. Ventilation was performed at 120 breaths per minute, a stroke volume of 200 μl, and a positive-end expiratory pressure (PEEP) of 3 cm $H_2O$. Mice were laid in the lateral decubitus position on the right side and the thorax was opened for direct access to the lung [12]. A thoracic suction window with a 12 mm coverslip (#1.5) was placed in a custom-built microscopy stage holder and the lung was restrained from moving with a vacuum of around 130 mm Hg. The mouse stage was turned upside down and placed on an inverted Nikon A1plus microscope on a 40x water-immersion objective (NA 1.15) with a resonance scanner. After focusing on a lung region, a z-stack was acquired without particles. Next, the lung was ventilated with pure oxygen for several minutes, ventilation stopped for 60 to 90 seconds until the alveolar space collapsed and particles (3.8 μg/g body weight particles, Table 1) in perfluorocarbon FC-43 (PFC) instilled through the PE-10 tubing directly into the lung.

The particles in PFC were prepared immediately before instillation as follows: an aqueous suspension of Ø 50 nm red fluorescent silica particle was emulsified in PFC by three consecutive sonication of 30 seconds each on ice, (Sonic Ruptor 250, Omni International, USA).

Z-Stacks were acquired at the indicated time points; all images were taken with Nikon NIS-Elements AR 4.20. Calculations were performed with ImageJ 1.50e and the "analyze particles" plugin. Stacks of images were merged for maximum intensity Z projection. The threshold was adjusted for the images prior to, and after, particle exposure according to Otsu [13], and the numbers and sizes of the agglomerates were analyzed. After background subtraction, the number and sizes of agglomerates were calculated following the initial settling at 2 minutes and every 10 minutes thereafter. For single plane analysis Renyi's Entropy threshold was applied before analyzing.

### *In vivo* preparation for transmission electron microscopy

Mice were anesthetized using 5% isoflurane and fluorescent amorphous silica particles of 50 nm or silica nanoparticles with a size of 58 nm and silver core of 28 nm were instilled into the trachea at concentrations described in Table 1. After 1 and 24 h, mice were euthanized with 22 mg/kg pentobarbital (Ceva, France). Samples of the lower left lobe of the lung were fixed in 2.5% gluteraldehyde, 1.6% formaldehyde in 0.1 M cacodylate buffer, pH 7.4 overnight. For $TiO_2$ samples were post-fixed in 1% osmium tetroxide buffered with cacodylate for 1 h for later analysis by transmission electron microscopy (TEM). After polymerization 70 nm thin vertical sections were cut from a representative area with cells (Ultracut E, Reichert-Jung, Vienna, Austria). Sections were stained with aqueous uranyl acetate and Reynolds's lead citrate and observed under a Hitachi H-7650 TEM (Hitachi) at 80 kV with an AMT16000 digital camera (Advanced Microscopy Techniques, USA).

### Whole organ imaging

Far-red fluorescent particles were instilled at 36 μg/g body weight. After 24 h liver, spleen and kidneys were collected. Images were acquired in a small animal imaging system (In-Vivo Xtreme 4MP, USA) at an excitation and emission wavelength of 730 and 790 nm, respectively and with a 5 seconds exposure time.

### *In vitro* uptake studies

A549 (ATCC, Virginia, USA) were maintained in F12 (Gibco, USA) supplemented with 10% of fetal bovine serum (Gibco, USA), 2 mM L-Glutamine, and 10 mM Hepes (Life Technologies, Canada). 2–2.5 μg of 50 nm red fluorescent nanoparticles were added to a confluent

monolayer of A549 in a 35 mm ibidi bottom dish for 1 h and then washed to remove unbound nanoparticles. Samples were then fixed and visualized by an ELYRA PS.1 inverted confocal microscope equipped with 63x 1.4NA objective (Zeiss, Germany). Stacks were recorded 0.3 μm apart from apical side to basolateral side. A representative plane near to the middle of the stack was chosen for mean fluorescence intensity (MFI) calculation and to ensure the particle was within the cell. By using imageJ MFI was calculated from five arbitrary areas within every cell and a minimum of 20 cells were counted to obtain mean fluorescence intensity/unit area (MFI/μm$^2$). A minimum of four experiments were performed. Alternatively, we have also measured total fluorescence from a maximum intensity projection of all the optical sections. Both measurement strategies showed a similar result. Particle uptake for cells at the middle of the stack was quantified using the imageJ analyze particle plug-in.

## Data and statistical analysis

All experiments were performed either in triplicate or as indicated in the figure legends. The results are described as mean ± standard deviation. Statistical analysis was performed by student's t-test and, where applicable, p-values were adjusted according to Bonferroni for multiple comparisons. Significant levels are indicated as * $p < 0.05$, ** $p < 0.01$, *** $p < 0.001$, and **** $p < 0.0001$. Statistical calculations and graphs were performed using either GraphPad Prism (version 5.0) or R software (version 3.5.1) [14] and the plot3D (version 1.1.1) package [15].

## Results

To investigate particle clearance from the alveolar region in situ, in real time, we employed intravital microscopy. We used perfluorocarbon (PFC) to homogenously deliver dispersed nanoparticles to the peripheral lung. PFC also improved the microscopy (see "discussion"). During the microscopy, the animals were subjected to partial-liquid ventilation, a procedure that is not harmful [16,17].

For the microscopy, z-stacks of 60-μm depth were acquired before and directly after instillation of bright, exceptionally stable fluorescent nanoparticles (S1H Fig). Z-stacks were then acquird in intervals of at most ten minutes for one hour. The alveolar walls were easily identified due to their strong auto-fluorescent at 488 nm (maximum projection image, Fig 1A), a property previously exploited for lung imaging [18]. The instilled particles settled on the alveolar wall in agglomerates of variable size and shape (Fig 1C and 1D). Smaller agglomerates disappeared one-by-one from view whereas larger entities were gradually reduced in size (Fig 1E and 1F). For instance, arrows in Fig 1D indicate agglomerates that vanished or diminished in Fig 1F. Quantifying the number and area of agglomerates in the stack showed a reduction to about half after 30 minutes (Fig 2C), consistent with the published biokinetics for the initial phase of clearance [19]. A fraction of particle agglomerates remained stationary within the observation timeframe of one hour (Fig 2C), reflecting a second, slower phase of clearance, also found by Kreyling et al. [19].

Next, we analyzed the z-stacks for movement of particles towards alveolar ducts over time (i.e. we analyzed whether particles agglomerates changed position within the alveolar lumen over the time of observation). This was not observed within the experimental timeframe (Fig 2A and 2B). Rather the agglomerates did not change their location when becoming reduced in size or before disappearing. We therefore hypothesized that the rapid phase of particle removal from the alveolar lumen occurs by crossing of the epithelium at the site of where particles have settled. Before investigating this hypothesis in mice, we tested as a control *in vitro* whether exposure to PFC affects particle interaction with the epithelium. A confluent layer of A549

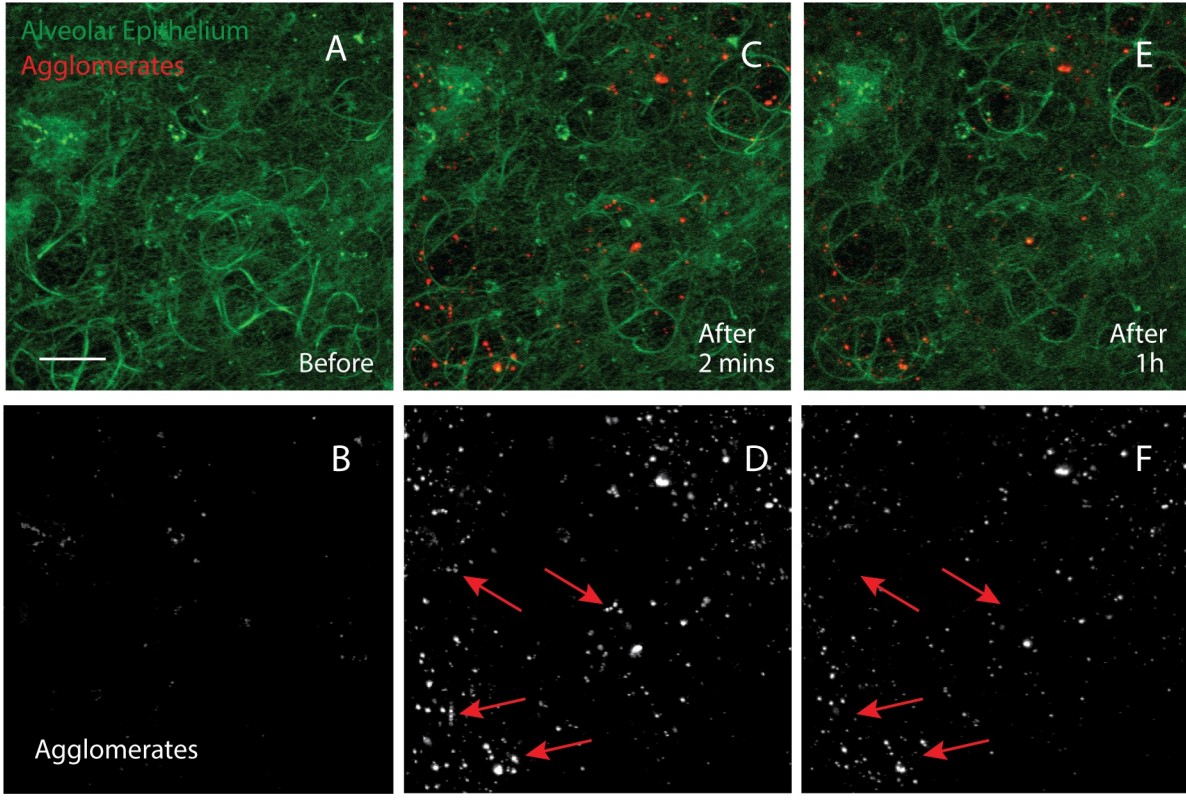

**Fig 1. Intravital lung imaging of silica nanoparticles in mice. A-F:** intravital microscopy of an alveolar region prior to instillation (A), 2 minutes after (C) and 1 h (E) after instilling Ø 50 nm red fluorescent amorphous silica particles, suspended in PFC (maximum intensity projection of a 60 μm-deep stack). The alveolar wall is auto-fluorescent in green. The particles are in agglomerates of variable size, similar to those observed by TEM (Fig 3). (B), (D), and (F) show the red channel only for better clarity. Agglomerates either have shrunk in size and intensity one hour after instillation, or have vanished entirely (red arrows). Note that the particles used do not photo bleach under the illumination chosen (S1H Fig). space bar: A-F = 20 μm shown in A.

alveolar epithelial cells was exposed to nanoparticles either dispersed in PFC or in media under fluorescent microscopy (S2A Fig). In both cases, the response of the cells to the particles was rapid endocytosis, with no significant difference between the two exposure types (S2B Fig). Further, we did not observe an increase of apoptotic cells under the microscope in our *in vitro* experiments.

To see if particles removed from the lung had become systemic, In-Vivo Xtreme 4MP whole-animal imaging was conducted. Results with entire animals were not achieved because of a high background against a low signal. However, after examining isolated organs, we found particle accumulation in the liver and kidneys 24 h after instillation (Fig 2D). This indicates that particles had become systemic. Renal- and hepatobiliary clearance has been shown for systemic silica nanoparticles, explaining their accumulation in the respective organs [6,20].

To investigate the alveolar epithelial crossing, we instilled particles in buffer suspension into the lungs of mice, euthanized them at either 1 h or 24 h after exposure, and prepared the lower left lobes of the lungs for transmission electron microscopy (TEM). We used a range of different particles and concentrations (Table 1). Following the 1 h exposure, nanoparticles were found in various stages of crossing the epithelium. The particles were mostly in agglomerates as described by Liu et al. [11] (see also S1E, S1F and S1G Fig). Agglomerates adhered to the apical side of the alveolar epithelium (Fig 3A), or were in the process of being internalized (Fig 3B). Some particles were also found entangled in the tubular myelin of surfactant (Fig 3F).

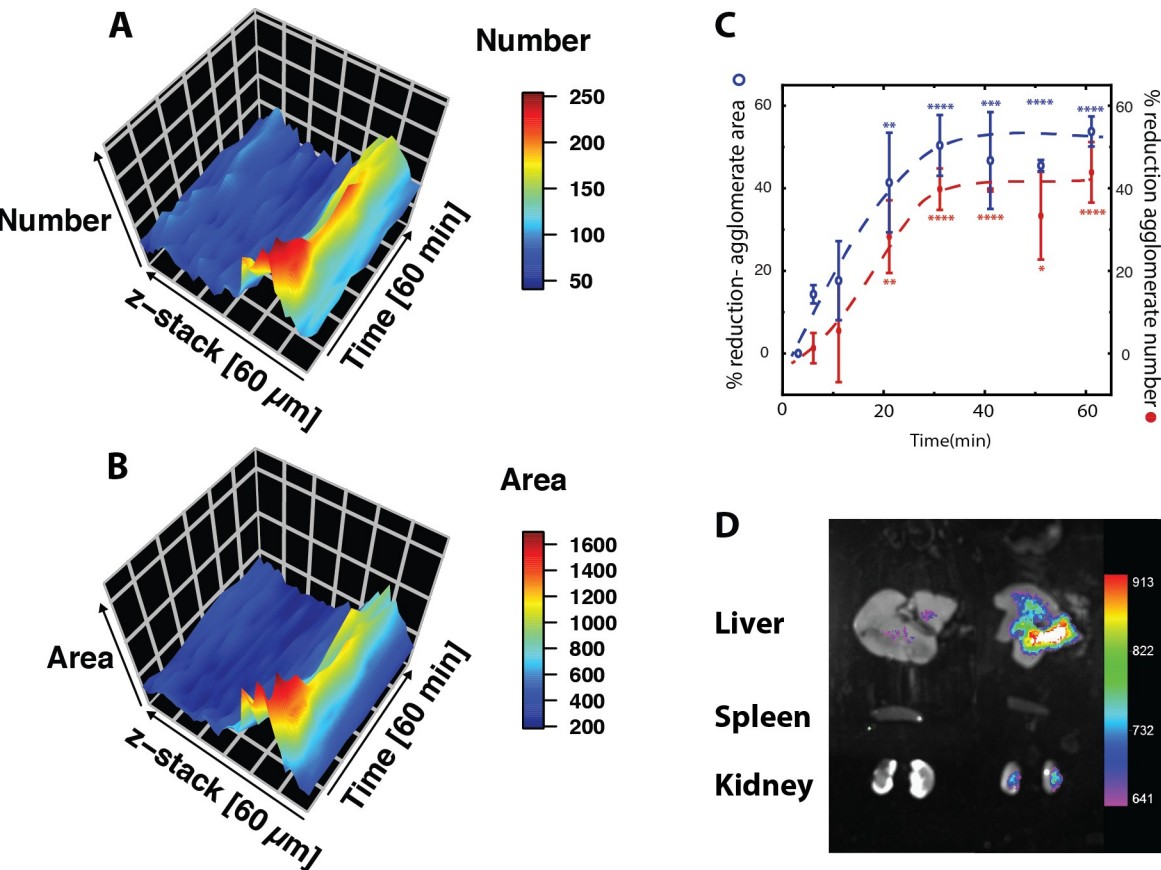

**Fig 2. Quantification of *in vivo* silica nanoparticle clearance.** Plot (z, t) of particles numbers (A) and area covered by particles (B) within the z-stack of Fig 1, over time. The stack starts with the alveolar wall close to the pleura (0 μm) and progresses towards the alveolar ducts (60 μm). Most particles are found close to the pleura (within alveoli). Particle number and area decrease within the 60 minutes time frame. The data indicates no movement from the alveoli towards the alveolar ducts, indicating that the particles do not leave the area via the airways. **C:** Quantification of particle clearance (data are mean ± SD, n = 3), with the reduction in agglomerate numbers in red and area in maximum intensity projection in blue. About half of the particles are cleared from the field of view over the first 30 minutes after exposure. The remainder of particles persist over the observation time of one hour. Significant changes are determined by student's t-test with Bonferroni adjusted p-values corrected for multiple comparisons (* p < 0.05, ** p < 0.01, *** p < 0.001, **** p < 0.0001). **D:** Whole organ imaging. Overlay of a white light image of the organs and a heat map-display of the fluorescence in liver (top), spleen (middle), and kidney (bottom) 24 h after instillation of Ø 50 nm Cy-7-labelled silica particles into the lung (right). The control images (left column) shows a low level of false positive counts in the liver.

Tubular myelin is the secretion form of pulmonary surfactant and is recognized by its fishnet appearance in TEM of lung sections. Agglomerates were also inside large, membrane-bound endosomal structures within type II epithelial cells (Fig 3C) and adherent to the apical- and basolateral side of alveolar type I cells (Fig 3D). However, they were not directly observed inside type I cells, possible because crossing of these very thin cells (of the order of a few times the diameter of the nanoparticles) is bound to be fast and unlikely to be observed by TEM. Finally, we observed particles in the lumen of pulmonary venules/arterioles (S3A and S3B Fig) indicating that they had crossed the endothelium to reach the blood stream. Another route for the particles to become systemic after crossing of the epithelium is via the lymph [7].

Interestingly, we also found alveolar macrophages with intracellular compartments of agglomerates (Fig 3F). Unlike epithelial cells, the agglomerates within macrophages were always associated with tubular myelin, suggesting that this might be required for macrophage phagocytosis (S4A and S4B Fig). Overall, macrophages were scarce within sections, and

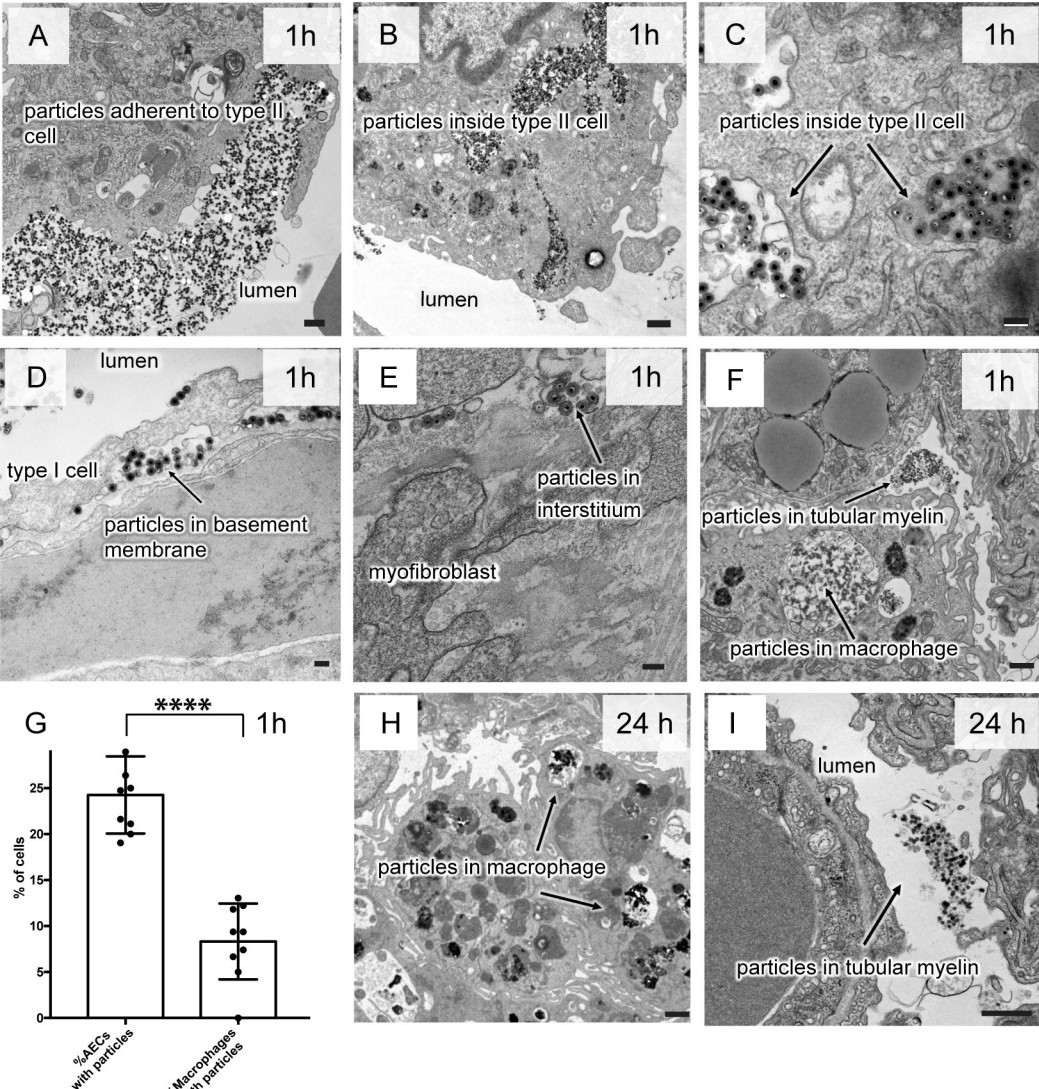

**Fig 3. Electron micrographs of lung sections after instillation of 50 nm-silica nanoparticles into the lungs of mice (3.8 µg/g body weight).** The micrographs show the progression of the nanoparticles from the alveolar lumen across the epithelium to the interstitial region or taken up by alveolar macrophages. Particles associated with- and inside epithelial cells (A-D) are in agglomerates, whereas particles that enter macrophages are entangled in tubular myelin (F, H). **A-G: Particle movement over the first hour of exposure.** A: Particle agglomerates adhere to type II cells. B: Particles have entered type II cells into large endosomes. C: The endosome containing the particles is surrounded by a membrane. For better visibility particles of 58 nm with silver core were used. D: Particles are found on both, the apical- and the basolateral side of the of type I epithelial cells. E: Particles have entered the interstitial region. F: Particle conglomerates are also taken up by alveolar macrophages. G: At the 1 h-timepoint, 25% of type II cells contained particles within regions of the lung that were exposed, whereas less than 10% of alveolar macrophages in the same region contained particles (data are mean ± SD, from three mice per condition and three sections per mouse. Significant changes are determined by student's t-test (**** p < 0.0001). **H, I particles after 24 h:** H: At this time point, we found no particles within alveolar epithelial cells. Particles were either in the alveolar lumen or I: inside alveolar macrophages. Particles inside macrophages and in the alveolar lumen were entangled in tubular myelin. Higher magnifications of uptake compartments with free agglomerates and with particles in a complex with tubular myelin are shown at high magnification in S4 Fig. space bars: A-B, F, H, I: 500 nm, C-E: 100 nm.

alveolar epithelial cells were the primary cells involved in processing of nanoparticles (Fig 3G). We did not observe an increase in macrophages or other inflammatory cells into the exposed lung regions or detect signs of edema or epithelial damage.

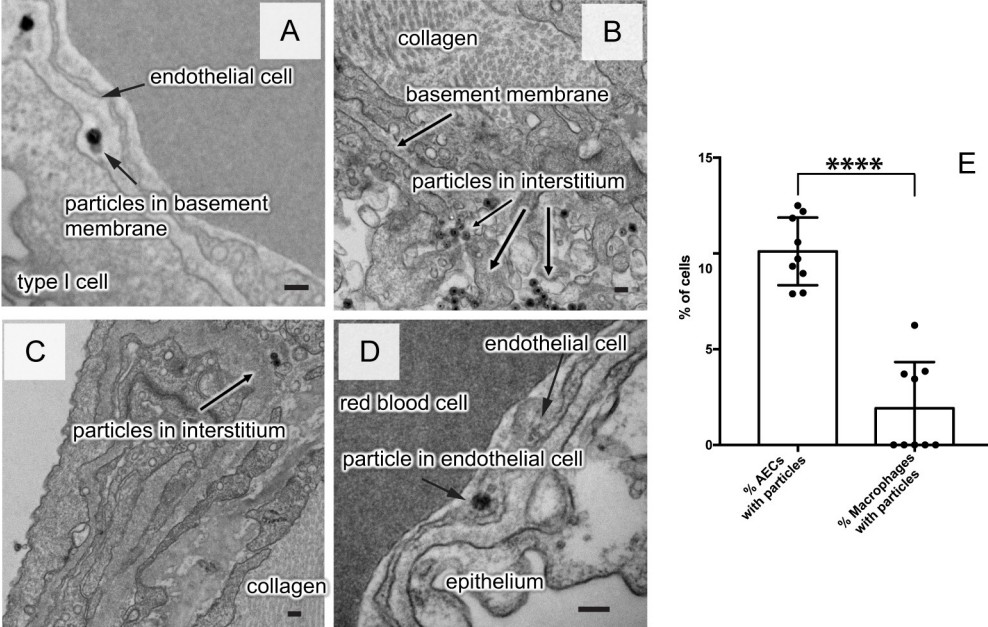

**Fig 4. Electron micrographs of low dose exposure (0.15 μg/g body weight) of silica nanoparticles in the lungs of mice.** Because particles were now individual rather that in agglomerates, we used ø 58 nm silica nanoparticle with 28 nm silver core for better visibility. After 1h, the lung was fixed and the left lower lobe sectioned for visualization. **A:** Particles between the epithelium and the endothelium, **B,C:** within the interstitium, **D:** inside a microvascular endothelial cell. **E:** Percentage of the alveolar epithelial cells (AECs) and of alveolar macrophages within TEM sections of mouse lungs that contained particles (0.15 μg/g body weight). Data are mean ± SD, from three mice per condition and three sections per mouse. Significant changes are determined by student's t-test (**** p < 0.0001). space bars: A-D = 100 nm.

In conclusion, trans-epithelial clearance is an effective and fast path for silica nanoparticles to be removed from the lung. The process does not appear to cause lung injury.

To investigate the fate of particles that had not accessed the rapid transepithelial pathway, we exposed mice as described above and sacrificed them after 24 h. At this point, the particles overall were relatively scarce, as compared to the 1 h-time point. Remaining particles in the alveolar lumen were associated with tubular myelin. Size and morphology of the particles were no different for the 1 h- and the 24 h-timepoints, indicating that particles themselves were stable. There were no particles adherent to- or within epithelial cells (Fig 3I). Macrophages contained agglomerates of particles within large endocytic compartments (Fig 3H). These particles were always associated with tubular myelin.

For the TEM experiments, we used moderate to low particle concentrations as compared to reported mouse silica particle instillation experiments [21] (3.0–3.8 μg/g body weight, comparable to intravital microscopy). In addition, a dose ~20 times lower (0.15 μg/g body weight) was used to determine if the dose affects the nature of the clearance pathway. At this low dose, particles were more often found individually rather than in agglomerates. They were few and far between in TEM sections. Clearance appeared to follow the same path to the agglomerates formed at higher concentrations (Fig 4A–4D).

To investigate if this pathway is relevant to other particle chemistries, we instilled TiO$_2$ nanoparticles of < 100 nm (1% Mn-doped). TiO$_2$ nanoparticles have an overall low toxicity [22]. Histology revealed a similar transport process across the lung epithelium as for amorphous silica (S3C Fig and S5A and S5B Fig).

In summary, amorphous silica and $TiO_2$ are cleared from the peripheral lung by rapidly crossing the alveolar epithelium. Alveolar macrophages internalized particles associated with tubular myelin.

## Discussion

Inhaled nanoparticles are predominantly deposited in the alveolar region [2,23]. The current publication shows for silica and $TiO_2$ that nanoparticles may effectively be transcytosed across the alveolar epithelium to be cleared from the lung. We found that particles predominantly crossed as agglomerates which may have contributed to the efficiency of this pathway. This pathway has been predicted for the particle types used here as well as many other nanoparticles. Published biokinetics for noncationic synthetic nanoparticles, $ZnO_2$-, $TiO_2$- and silica-, among other nanoparticles indicates rapid and effective translocation from the alveolar lumen into the bloodstream [6,7]. Choi, et al [7] state:"...there is rapid transepithelial translocation of nanoparticles from the alveolar luminal surface into the septal interstitium, followed by quick translocation to the regional draining lymph nodes where further translocation into the bloodstream could occur...". A review by Oberdörster et al. [10] describes passage via the lymph to the lung periphery close to the pleura [24].

Regarding intravital lung microscopy of the particle exposure, the use of PFC proved important. Intravital lung microscopy suffers from excessive movement during ventilation and light scattering at the air-tissue interface. Another problem, the procedure limits the observable lung to a small region that cannot be changed once the microscopy has commenced. Sucessful imaging therefore depends on a homogenous and reliable deposition of particles within the field of view. Particle exposure via an aqueous bolus is not well suited for intravital microscopy because the distal region of the lung under observation is often shut off from the exposure by trapped air. Using a particle emulsion in PFC (i.e. an aqueous suspension of particles is emulsified in PFC) solved these problems. When instilled, the dispersion filled up the entire airspaces of the lower lung regions owing to its low viscosity, very low surface tension, high density (twice that of water), and immiscibility with water. In addition to reliably carrying the particles into the field of view, PFC improved the microscopy because it matches the refractive index of the alveolar lumen to the tissue to reduce light scattering. The motion of breathing was also strongly reduced and allowed us to observe nanoparticles over an extended period of time. We note that PFC preserves the airway and alveolar structure including the surfactant layer at the air-tissue interface [9], the thin layer of alveolar fluid and the epithelium, without causing injury or affecting gas exchange. PFC emulsions have been successful for delivery of emulsified nanoparticulate drugs to the lungs [25] [26].

What is the significance of the rapid transcytosis mechanism described in the current paper for lung homeostasis? The lung-air interface must be clear of foreign objects to ensure unobstructed air flow, gas exchange and prevent inflammation. The nasal cavity or the trachea-bronchial tree trap and remove larger particles. Nanoparticles behave increasingly like gas molecules with decreasing size and therefore bypass these defence mechanisms and settle where gas exchange occurs, in the alveolar lung. Clearance of these particles via the lymph and the blood proved highly effective for the particle types studied. Inhaled nanoparticles, while predominantly deposited in the alveolar region, may also settle in the bronchi and bronchioles. It will be interesting to see whether these particles are cleared via the mucociliary escalator like larger particles or traverse the (airway) epithelium. We note that the mucus with particles will mostly enter the gastrointestinal tract from where they potentially also can become systemic.

Not all nanoparticles follow the pathway described here and will be the subject of future studies. For example, carbon particles from coal mining or cigarette smoke [27] may

accumulate within the alveolar lumen resulting in emphysema. Particles that accumulate in the interstitium may induce pneumoconiosis [28]. As well, not all particles that reach the bloodstream will be cleared without causing damage, but be responsible for cardiovascular, renal or neural effects [29]. Another subject that warrants more research is the detailed cell-biology of the transcytosis mechanism.

In conclusion, the current paper contributes an essential pathway that aligns with the reported kinetics of lung clearance of a broad spectrum of nanoparticles [5–7]. In addition to the implications for nanoparticle toxicicty, understanding of the transepithelial pathway may may be important for the design of inhaled particulate drugs, broadening the spectrum of drugs that may be delivered via the lung. Therapeutic substances may include new peptide and protein drugs, where delivery via the lungs is particularly important because of the many limitations of other delivery modes [30–32].

## Supporting information

**S1 Fig. Nanoparticle characterization. A:** TEM of agglomerates of ø 50 nm red fluorescently labeled silica nanoparticle used for the intravital microscopy and **B:** ø 58 nm silica nanoparticle with 28 nm silver core used for electron microscopy of mouse lungs exposed at a low dose. **C:** ø 50 nm Cy7 infrared labeled silica nanoparticles used for isolated organ imaging. **D:** 2.5 μg of red silica nanoparticles were suspended in 2 ml culture media and the agglomerates were imaged using confocal microscopy. **E-F**: Hydrodynamic radii of agglomerates of ø 50 nm silica nanoparticle dispersed in phosphate-buffered saline (PBS) at 37˚C, immediately following sonication (0 minutes) and after 15 minutes at 50 μg/mL (E) and 250 μg/mL (F), showing agglomeration is time and concentration dependent. **G:** Hydrodynamic radii of agglomerates of ø 50 nm-nanoparticle dispersed in phosphate-buffered saline (PBS) at 37˚C (black) and cell culture media (blue). Note that the media drives agglomeration. **H:** *In vitro*, the red fluorescent nanoparticles used for the intravital microscopy do not bleach for a similar laser exposure as used for the intravital microscopy. For comparison, a FITC-labelled particle bleaches upon a similar exposure, as shown in the reduction of mean fluorescence intensity (MFI).
(TIF)

**S2 Fig. *In vitro* nanoparticle uptake in perfluorocarbon. A:** Sketch of the experimental setup to study the effect of delivering nanoparticles in perfluorocarbon (PFC) to lung epithelial (A549) cells on uptake as compared to delivery of particles in media. A549 cells cultured on 35 mm dishes and exposed to nanoparticles in media (Top) or suspended in PFC (bottom). **B:** Quantification of fluorescence (MFI) after 1h nanoparticle uptake in media (control), media + $NaN_3$ + 2-deoxy-D-glucose (inhibition of endocytosis), PFC, PFC & $NaN_3$ + 2-deoxy-D-glucose.
(TIF)

**S3 Fig. Silica and $TiO_2$ nanoparticles transported in the bloodstream. A-B:** Mice were instilled with Cy7, infrared nanoparticles that have been used for mouse whole organ imaging. **B: enlarged view of A showing** evidence of particles in the plasma of the bloodstream. **C:** showing an example of a blood cell with a titanium oxide particle space bars: A & C = 500 nm, B = 100 nm
(TIF)

**S4 Fig. Particle uptake in alveolar epithelial cells vs. macrophages. A:** Magnification from Fig 3B, Electron micrographs of lung sections after instillation of 50 nm-silica nanoparticles into the lungs of mice. The particles inside an uptake compartment of type II alveolar epithelial cells are in dense agglomerates, similar to S1A Fig. The particles do not share the compartment

with membranous structures. **B:** Magnification from Fig 3F. Particles inside uptake compartments of macrophages are dispersed and associated with the membranous structures of tubular myelin. **C:** Tubular myelin with entangled particles in the alveolar lumen looks like the tubular myelin with particles inside macrophages. space bars: 500 nm
(TIF)

**S5 Fig. Titanium oxide nanoparticles cross the epithelium-like silica nanoparticles. A-B:** Mice (n = 2) were instilled with 3.8 μg/g **body weight** 1% Mn-doped $TiO_2$ (ø<100 nm particles). **A:** shows $TiO_2$ nanoparticles inside a Type II cell. **B:** shows an example of $TiO_2$ nanoparticles having crossed the alveolar epithelium similar to the silica nanoparticles. space bars: A = 500 nm, B = 100 nm.
(TIF)

## Acknowledgments

We thank W. Dong for preparation of TEM sections, K. Wojcik for helping to establish intravital microscopy, L. Gunasekara, and A. Yang for assistance in mouse experiments.

## Author Contributions

**Conceptualization:** Pascal Detampel, Anutosh Ganguly, Matthias W. Amrein.

**Data curation:** Anutosh Ganguly.

**Formal analysis:** Pascal Detampel, Anutosh Ganguly.

**Funding acquisition:** Pascal Detampel, Sara Tehranian, Matthias W. Amrein.

**Investigation:** Pascal Detampel, Anutosh Ganguly, Sara Tehranian, Matthias W. Amrein.

**Methodology:** Anutosh Ganguly, Santiswarup Singha, Björn Petri.

**Project administration:** Anutosh Ganguly, Matthias W. Amrein.

**Resources:** Santiswarup Singha, Pere Santamaria, Ayodeji A. Jeje, Clifford S. Cho.

**Supervision:** Anutosh Ganguly, Ayodeji A. Jeje, Matthias W. Amrein.

**Validation:** Anutosh Ganguly, Francis Green, Clifford S. Cho, Matthias W. Amrein.

**Writing – original draft:** Pascal Detampel, Matthias W. Amrein.

**Writing – review & editing:** Pascal Detampel, Anutosh Ganguly, Francis Green, Clifford S. Cho, Matthias W. Amrein.

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
