## [Decision Letter · Decision Letter 0]

17 Jul 2019

PONE-D-19-17546

In vivo clearance of nanoparticles by transcytosis across alveolar epithelial cells

PLOS ONE

Dear Dr. Ganguly,

Thank you for submitting your manuscript to PLOS ONE. After careful consideration, we feel that it has merit but does not fully meet PLOS ONE’s publication criteria as it currently stands. Therefore, we invite you to submit a revised version of the manuscript that addresses the points raised during the review process.

We would appreciate receiving your revised manuscript by Aug 31 2019 11:59PM. To enhance the reproducibility of your results, we recommend that if applicable you deposit your laboratory protocols in protocols.io, where a protocol can be assigned its own identifier (DOI) such that it can be cited independently in the future. For instructions see: http://journals.plos.org/plosone/s/submission-guidelines#loc-laboratory-protocols

We look forward to receiving your revised manuscript.

Kind regards,

Salik Hussain, D.V.M, M.S., Ph.D.,

Academic Editor

PLOS ONE

Journal Requirements:

2.Thank you for stating the following in your Competing Interests section: "None"

Reviewers' comments:

Reviewer's Responses to Questions

**Comments to the Author**

1. Is the manuscript technically sound, and do the data support the conclusions?

Reviewer #1: Yes

Reviewer #2: Yes

Reviewer #3: Partly

2. Has the statistical analysis been performed appropriately and rigorously? 

Reviewer #1: No

Reviewer #2: Yes

Reviewer #3: Yes

3. Have the authors made all data underlying the findings in their manuscript fully available?

Reviewer #1: Yes

Reviewer #2: Yes

Reviewer #3: Yes

4. Is the manuscript presented in an intelligible fashion and written in standard English?

Reviewer #1: Yes

Reviewer #2: Yes

Reviewer #3: Yes

5. Review Comments to the Author

Reviewer #1: Ganguly et al. investigated the transcytosis of silica nanoparticles across alveolar epithelial cells. To perform these experiments they used a novel combination of techniques including intravital microscopy, TEM, whole organ imaging, and in vitro studies. The modification to inverted lung microscopy to include partial liquid ventilation to image the nanoparticles is interesting from a technical/experimental point of view. Also the use of relatively inert silica nanoparticles that have their kinetic profiles already established was a strength of the study and removed some factors such as inflammation, edema, and other toxicity that may have occurred with use of another particle type. The use of TiO2 nanoparticles was able to validate their findings from silica exposures.

Comments

1. The materials sections states that 4 silica nanoparticle sizes were purchased. It is unclear in specific experiments which nanoparticle was utilized. For example, in the in vitro assessment of A549 cell internalization it only states the concentration not the size. Although mentioned within the materials the 30 and 500 nm nanoparticles do not appear to have been used.

2. The characterization of the nanoparticles is sufficient and well done.

3. The results as written include a large amount of discussion. These portions need to be moved to the discussion. As currently written the discussion is lacking in depth.

4. Figures 2C, 3G, and 4E include graphs with error bars. These figures lack statistical analysis. Please perform statistical assessments of these data to establish if differences are significant. Also add a section into the methods detailing statistical information such as group sizes, statistical tests, and threshold of significance.

Reviewer #2: The manuscript “In vivo clearance of nanoparticles by transcytosis across alveolar epithelial cells” by Ganguly, et al, is well-written and describes research I believe would be of interest to the readership of PLOS One. The mechanisms by which nanoscale particles move from the alveolar space to systemic circulation is not fully understood and this manuscript provides impactful insight into this topic.

I have some minor revisions I would recommend before publication in PLOS One:

1) While the hydrodynamic diameter of the nanoparticle agglomerates formed in this study was measured for PBS and culture medium, this agglomeration size is missing for the perfluorocarbon vehicle used in the partial liquid ventilation. Given the importance of this portion of the manuscript to the conclusions being drawn, it is critical that some measure of agglomeration in PFC is made. It is not described in the methods section what device was used to measure these diameters, but the Malvern Zetasizer can also be used for organic solvents.

2) The manuscript does not make a convincing case that significant amounts of nanomaterial are not being caught/cleared by the mucociliary escalator, especially since this method of clearance is widely established for micro-scale/fine particles. The assertion that “the particles do not leave the area via the airways” relies on the 60um z-stack in figure 2, suggesting that there are more particles in the alveolar sacs than in the ducts. However, large percentages of particle agglomerates should be getting caught in the conducting airways before they reach the alveolar ducts (as this is their purpose), and this effect would be even more pronounced in human lungs where there is more branching. The manuscript should include some discussion on the relative proportions of inhaled nanoparticle agglomerates which reach the alveolar space vs that which is cleared by the conducting airways.

3) Related to the previous point, the clearance of particles by mucociliary action would put them into the gastrointestinal tract, providing a new potential point of entry into systemic circulation. While it is unlikely that the GI tracts of the study mice are still available for analysis, some discussion into why the authors believe that alveolar uptake is a more important contributor to systemic nanoparticle circulation than GI uptake would be warranted.

4) Nanosilica is known to be cytotoxic and causes redox stress in most other in vitro research. It is recommended that, at the least, some measure of cytotoxicity at study doses be provided in the supplementary information. Some discussion on the justification for using the doses applied in this study (as in, comparisons to “real world” exposures) as well as why no cytotoxicity or inflammation was observed (what doses would cause these effects and are they realistic exposures?) would be helpful to establish the relevance of the manuscript.

5) More speculatively, it is surprising to me that a “fast track” mechanism for transporting inhaled particles in bulk to the bloodstream/lymph would even exist, given how many other safeguards are in place to prevent exactly that. I am curious to hear the author’s opinions on why such a mechanism might exist and what purpose this would serve the animal.

Reviewer #3: The authors studied the mechanism of silica nanoparticles (50 nm) to get cleared by crossing the lung epithelium within 30 minutes. The authors presented fluorescence data and TEM showing the particles in transcytosis. This is an interesting observation, however, more proof is needed on this clearance pathway.

1. Amorphous Silica has been shown to be soluble in the presence of salts and proteins, the authors should test the dissolution rate of the these nanoparticles in physiologically relevant lung fluid. This is important because fluorescent labels may be released to cause an artifact of particle clearance.

2. More quantitative method of clearance is the ICP-MS, especially for the silica with silver cores, which could give more accurate data than fluorescence.

3. Transcytosis is often mentioned, but rarely a detailed pathway is described. This study lacks a mechanistic understanding other than an interesting phenomenon.

6. PLOS authors have the option to publish the peer review history of their article (what does this mean?). If published, this will include your full peer review and any attached files.

Reviewer #1: No

Reviewer #2: No

Reviewer #3: No

---

## [Author Response · Author response to Decision Letter 0]

31 Aug 2019

Answers to reviewers: We are grateful for the insightful criticism by the reviewers and hope that we have been able to address the problems.

---

Reviewer #1: Ganguly et al. investigated the transcytosis of silica nanoparticles across alveolar epithelial cells. To perform these experiments they used a novel combination of techniques including intravital microscopy, TEM, whole organ imaging, and in vitro studies. The modification to inverted lung microscopy to include partial liquid ventilation to image the nanoparticles is interesting from a technical/experimental point of view. Also the use of relatively inert silica nanoparticles that have their kinetic profiles already established was a strength of the study and removed some factors such as inflammation, edema, and other toxicity that may have occurred with use of another particle type. The use of TiO2 nanoparticles was able to validate their findings from silica exposures.

Comments

1. The materials sections states that 4 silica nanoparticle sizes were purchased. It is unclear in specific experiments which nanoparticle was utilized. For example, in the in vitro assessment of A549 cell internalization it only states the concentration not the size. Although mentioned within the materials the 30 and 500 nm nanoparticles do not appear to have been used.

We thank the reviewer for this important question. Indeed, we did not use the 30 and 500 nm nanoparticles in this set of experiments and we are sorry for this confusion. The material and methods section have been updated accordingly.

2. The characterization of the nanoparticles is sufficient and well done.

Thank you.

3. The results as written include a large amount of discussion. These portions need to be moved to the discussion. As currently written the discussion is lacking in depth.

We agree with the reviewer. We moved most of the discussion from the results- to the discussion section. We left some discussion in the results section to provide the rationale for our experimental approach.

4. Figures 2C, 3G, and 4E include graphs with error bars. These figures lack statistical analysis. Please perform statistical assessments of these data to establish if differences are significant. Also add a section into the methods detailing statistical information such as group sizes, statistical tests, and threshold of significance.

We thank the reviewer for this comment. We now indicate statistical significance in figures 2C, 3G, and 4E and added a section into the methods describing statistical tests. It can be noted that in Figures 3G and 4E, as well as at the later time points also for figure 2C (with one exception), all differences are highly significant (p-values are < 0.001 or < 0.0001).

---

Reviewer #2: The manuscript “In vivo clearance of nanoparticles by transcytosis across alveolar epithelial cells” by Ganguly, et al, is well-written and describes research I believe would be of interest to the readership of PLOS One. The mechanisms by which nanoscale particles move from the alveolar space to systemic circulation is not fully understood and this manuscript provides impactful insight into this topic.

I have some minor revisions I would recommend before publication in PLOS One:

1) While the hydrodynamic diameter of the nanoparticle agglomerates formed in this study was measured for PBS and culture medium, this agglomeration size is missing for the perfluorocarbon vehicle used in the partial liquid ventilation. Given the importance of this portion of the manuscript to the conclusions being drawn, it is critical that some measure of agglomeration in PFC is made. It is not described in the methods section what device was used to measure these diameters, but the Malvern Zetasizer can also be used for organic solvents.

We thank the reviewer for this important comment and suggestion. For intravital microscopy, an aqueous suspension of nanoparticles was emulsified in PFC. This means that the particles remained within microscopic droplets of buffer. Hence, we assume that the agglomeration is no different than for particles in aqueous suspension. As well, we note that the agglomerates are directly observed in the intravital microscope and seem to correlate with the observations made in TEM. Finally, we note that emulsion of water in PFC will scatter in DLS measurements (Malvern Zetasizer). The Zetasizer has indeed been used in the past to determine droplet size of water in PFC (Grapentin C, Barnert S, Schubert R. Monitoring the Stability of Perfluorocarbon Nanoemulsions by Cryo-TEM Image Analysis and Dynamic Light Scattering. PLOS ONE. 2015;10: e0130674. doi:10.1371/journal.pone.0130674). However, we realized that we need to clarify our methods section regarding the preparation of the emulsion as follows:

“The particles in PFC were prepared immediately before instillation as follows: an aqueous suspension of Ø 50 nm red fluorescent silica particle was emulsified in PFC by three consecutive sonication of 30 seconds each on ice, (Sonic Ruptor 250, Omni International, USA).”

Further, we have revised our manuscript to note this limitation in the Results.

2) The manuscript does not make a convincing case that significant amounts of nanomaterial are not being caught/cleared by the mucociliary escalator, especially since this method of clearance is widely established for micro-scale/fine particles. The assertion that “the particles do not leave the area via the airways” relies on the 60um z-stack in figure 2, suggesting that there are more particles in the alveolar sacs than in the ducts. However, large percentages of particle agglomerates should be getting caught in the conducting airways before they reach the alveolar ducts (as this is their purpose), and this effect would be even more pronounced in human lungs where there is more branching. The manuscript should include some discussion on the relative proportions of inhaled nanoparticle agglomerates which reach the alveolar space vs that which is cleared by the conducting airways.

We thank the reviewer for this important set of insights. We have accommodated these suggestions by expanding the Discussion of the manuscript to better emphasize why the pathway described is important. We now point out the limitations of the current study and the problems not addressed by us.

In direct answer to the reviewer, we argue that we address an important aspect of particle clearance in the current submission. We agree that retention and clearance of particles deposited in the nasal region and the upper airways is well established for micron-sized and larger particles. They are either kept from entering the lung or are removed by the mucociliary pathway. We also agree with the reviewer that it will be interesting to see what happens to nanoparticles that deposit in this region (i.e. the degree to which they are removed via the mucociliary pathway or whether they follow a different path). However, over 50% of nanoparticles deposit in the alveolar region, depending on size (reviewed e.g. Oberdürster, Günter. "Toxicology of ultrafine particles: in vivo studies. "Philosophical Transactions of the Royal Society of London. Series A: Mathematical, Physical and Engineering Sciences 358.1775 (2000): 2719-2740.). 

We clarify this point now in the manuscript: “Inhaled nanoparticles, while predominantly deposited in the alveolar region, may also settle in the bronchi and bronchioles. It will be interesting to see whether these particles are cleared via the mucociliary escalator or traverse the (airway) epithelium.”

Particles similar to the ones studied here were shown to enter the bloodstream from this region with a half-life time of about 0.5 h and were almost fully removed from the lung within 24 h (cited and discussed by us). This is incompatible with the canonical clearance mechanism for alveolar “dust” via alveolar macrophages and calls for an alternative pathway. We now show that this pathway involves crossing of the alveolar epithelium.

Regarding the 60-um z-stacks, we agree that these results do not address what happens to particles that have deposited in the airways. Rather, observations of the 60-um z-stacks indicate that particles that have deposited on the alveolar wall do not migrate towards the bronchioles. We thus exclude that the alveolar population of particles leaves via the airways. We write this now more clearly: “Next, we analyzed the z-stacks for movement of particles towards alveolar ducts over time (i.e. we analyzed whether particles agglomerates changed position within the alveolar lumen over the time of observation).”

3) Related to the previous point, the clearance of particles by mucociliary action would put them into the gastrointestinal tract, providing a new potential point of entry into systemic circulation. While it is unlikely that the GI tracts of the study mice are still available for analysis, some discussion into why the authors believe that alveolar uptake is a more important contributor to systemic nanoparticle circulation than GI uptake would be warranted.

We have addressed this important recommendation by discussing this point in the revised manuscript as follows: “We note that the mucus with particles will mostly enter the gastrointestinal tract from where they potentially also can become systemic.”

4) Nanosilica is known to be cytotoxic and causes redox stress in most other in vitro research. It is recommended that, at the least, some measure of cytotoxicity at study doses be provided in the supplementary information. Some discussion on the justification for using the doses applied in this study (as in, comparisons to “real world” exposures) as well as why no cytotoxicity or inflammation was observed (what doses would cause these effects and are they realistic exposures?) would be helpful to establish the relevance of the manuscript.

We thank the reviewer for this important question. As indicated in our manuscript, we did not observe an increase in macrophages or other inflammatory cells into the exposed lung regions or detect signs of edema or epithelial damage. Further, we never observed any apoptotic processes with these silica nanoparticles in our in vitro cell culture experiment using the alveolar epithelial A549 cells. We now state this observation in the Results:

“Further, we did not observe an increase of apoptotic cells under the microscope in our in vitro experiments.”

5) More speculatively, it is surprising to me that a “fast track” mechanism for transporting inhaled particles in bulk to the bloodstream/lymph would even exist, given how many other safeguards are in place to prevent exactly that. I am curious to hear the author’s opinions on why such a mechanism might exist and what purpose this would serve the animal.

This a fascinating question, and we have revised our manuscript to discuss this more specifically. We agree that larger inhaled particles are caught in the nasal cavity or removed via the mucociliary escalator. However, nanoparticles are preferentially deposited in the alveolar region where these defense mechanisms are not present. We argue that these particles are predominantly cleared from the lung via the bloodstream after crossing the alveolar epithelium, based on the biokinetics studies cited by us and our own study. We propose that this fast track mechanism is an important clearance mechanism for inhaled nanoparticles and possibly the main defense against some nanoparticles in the lung. We cite studies that show hepatic clearance or by urinary excretion and changed a section in the discussion to reflect this as follows:

“What is the significance of the rapid transcytosis mechanism described in the current paper for lung homeostasis? The lung-air interface must be clear of foreign objects to ensure unobstructed air flow, gas exchange and prevent inflammation. The nasal cavity or the trachea-bronchial tree trap and remove larger particles. Nanoparticles behave increasingly like gas molecules with decreasing size and therefore bypass these defence mechanisms and settle where gas exchange occurs, in the alveolar lung. Clearance of these particles via the lymph and the blood proved highly effective for the particle types studied. Inhaled nanoparticles, while predominantly deposited in the alveolar region, may also settle in the bronchi and bronchioles. It will be interesting to see whether these particles are cleared via the mucociliary escalator like larger particles or traverse the (airway) epithelium. We note that the mucus with particles will mostly enter the gastrointestinal tract from where they potentially also can become systemic.

Not all nanoparticles follow the pathway described here and will be the subject of future studies. For example, carbon particles from coal mining or cigarette smoke [27] may accumulate within the alveolar lumen resulting in emphysema. Particles that accumulate in the interstitium may induce pneumoconiosis [28]. As well, not all particles that reach the bloodstream will be cleared without causing damage, but be responsible for cardiovascular, renal or neural effects [29]. Another subject that warrants more research is the detailed cell-biology of the transcytosis mechanism.”

---

Reviewer #3: The authors studied the mechanism of silica nanoparticles (50 nm) to get cleared by crossing the lung epithelium within 30 minutes. The authors presented fluorescence data and TEM showing the particles in transcytosis. This is an interesting observation, however, more proof is needed on this clearance pathway.

1. Amorphous Silica has been shown to be soluble in the presence of salts and proteins, the authors should test the dissolution rate of the these nanoparticles in physiologically relevant lung fluid. This is important because fluorescent labels may be released to cause an artifact of particle clearance.

We thank the reviewer for this question. The fluorescent silica nanoparticles used for intravital microscopy contained the dye embedded covalently into the whole particle matrix, in contrast to surface-labeled spheres. We note that the particles remained stable over one hour with respect to the fluorescence signal in PBS (i.e. the duration of the intravital microscopy). This indicates that they do not photo bleach but also not dissolve either, at least not in PBS (Fig S1H). Furthermore, we note that the same particles were also used for TEM. We did not observe a noticeable change in size of these particles in vivo over time. When we compared TEM section from mice sacrificed 1 h after the exposure to sections from mice sacrificed at the 24 hours time point, we did not observe a difference in particle diameter (Figure 3). For an intuitive direct comparison, we here show an enlarged section of Fig 3F for 1-h time point and an enlarged section of Fig. 3 IH for the 24-h time point of the same particles at the same magnification. Therefore, we are confident that we can exclude relevant solubilisation of the used particles or leakage of dye from these particles during the investigated 24 hours. We state this now in the manuscript:

“Size and morphology of the particles were no different for the 1 h- and the 24 h-timepoints, indicating that particles themselves were stable.”

2. More quantitative method of clearance is the ICP-MS, especially for the silica with silver cores, which could give more accurate data than fluorescence.

We thank the reviewer for this question. We argue that a high quality standard of materials distribution within tissue is radio-labeling. This has already been done for the problem investigated and our reference to this work has been noted as a strength by another reviewer: “Also the use of relatively inert silica nanoparticles that have their kinetic profiles already established was a strength of the study and removed some factors such as inflammation, edema, and other toxicity that may have occurred with use of another particle type.” While we agree that ICP-MS could give us additional information, we are convinced that following the intact particles in the lung with the different microscopic approaches permitted us to draw the presented conclusion.

3. Transcytosis is often mentioned, but rarely a detailed pathway is described. This study lacks a mechanistic understanding other than an interesting phenomenon.

Thank you for this valuable comment. We agree that a comprehensive study of the cell biology of the transcytosis must be a next step. To address this point we have added the following section to the Discussion:

“Another subject that warrants more research is the detailed cell-biology of the transcytosis mechanism.”

---

## [Decision Letter · Decision Letter 1]

19 Sep 2019

In vivo clearance of nanoparticles by transcytosis across alveolar epithelial cells

PONE-D-19-17546R1

Dear Dr. Ganguly,

We are pleased to inform you that your manuscript has been judged scientifically suitable for publication and will be formally accepted for publication once it complies with all outstanding technical requirements.

With kind regards,

Salik Hussain, D.V.M, M.S., Ph.D.,

Academic Editor

PLOS ONE

Additional Editor Comments (optional):

Reviewers' comments:

Reviewer's Responses to Questions

**Comments to the Author**

1. If the authors have adequately addressed your comments raised in a previous round of review and you feel that this manuscript is now acceptable for publication, you may indicate that here to bypass the “Comments to the Author” section, enter your conflict of interest statement in the “Confidential to Editor” section, and submit your "Accept" recommendation.

Reviewer #1: All comments have been addressed

Reviewer #2: All comments have been addressed

Reviewer #3: All comments have been addressed

2. Is the manuscript technically sound, and do the data support the conclusions?

Reviewer #1: Yes

Reviewer #2: Yes

Reviewer #3: Yes

3. Has the statistical analysis been performed appropriately and rigorously? 

Reviewer #1: Yes

Reviewer #2: Yes

Reviewer #3: Yes

4. Have the authors made all data underlying the findings in their manuscript fully available?

Reviewer #1: Yes

Reviewer #2: Yes

Reviewer #3: Yes

5. Is the manuscript presented in an intelligible fashion and written in standard English?

Reviewer #1: Yes

Reviewer #2: Yes

Reviewer #3: Yes

6. Review Comments to the Author

Reviewer #1: The authors have adequately addressed reviewer feedback. Following revisions the manuscript is technically and statistically sound. This meets the criteria for publication in PLoS One.

Reviewer #2: The authors have addressed the minor revisions I requested, and I recommend publication of this work in PLOS ONE.

Reviewer #3: The authors sufficiently addressed the comments from reviewers, so now I would recommend its acceptance.

7. PLOS authors have the option to publish the peer review history of their article (what does this mean?). If published, this will include your full peer review and any attached files.

Reviewer #1: No

Reviewer #2: No

Reviewer #3: Yes: tian xia

---

## [Editor Report · Acceptance letter]

23 Sep 2019

PONE-D-19-17546R1 

In vivo clearance of nanoparticles by transcytosis across alveolar epithelial cells 

Dear Dr. Ganguly:

I am pleased to inform you that your manuscript has been deemed suitable for publication in PLOS ONE. Congratulations! Your manuscript is now with our production department. 

With kind regards,

on behalf of

Dr. Salik Hussain 

Academic Editor

PLOS ONE